# Recent Developments in the Hybridization of the Freeze-Drying Technique in Food Dehydration: A Review on Chemical and Sensory Qualities

**DOI:** 10.3390/foods12183437

**Published:** 2023-09-15

**Authors:** Chibuzo Stanley Nwankwo, Endurance Oghogho Okpomor, Nesa Dibagar, Marta Wodecki, Wiktor Zwierz, Adam Figiel

**Affiliations:** 1Department of Food Science and Technology, Federal University of Agriculture, Makurdi P.M.B 2373, Nigeria; toteupstar@outlook.com; 2International Centre for Biotechnology (ICB) Under the Auspices of UNESCO, University of Nigeria, Nsukka 410105, Nigeria; endyoghogho@gmail.com; 3Institute of Agricultural Engineering, Wroclaw University of Environmental and Life Sciences, 51-630 Wrocław, Poland; n.dibaga93@gmail.com; 4Veterinary Clinic for Small Animals Leverkusen, 51381 Leverkusen, Germany; mwodecki83@hotmail.com; 5Water Science and Technology Institute—H2O SCITECH, 51-351 Wrocław, Poland; wiktor.zwierz@f4ftech.com

**Keywords:** freeze-drying, hybrid freeze-drying, nutritive quality, sensory attributes

## Abstract

Freeze-drying is an excellent method for dehydration due to its benefits, including increased shelf-life, unique texture, and, in particular, good nutritive quality. However, the applicability of traditional freeze-drying systems in the food industry is still challenging owing to their prolonged drying duration, extraordinary energy usage, and high process cost. Therefore, the need to upgrade or develop conventional freeze-dryers for common or sophisticated food structures is ever-increasing. Enhancements to the freeze-drying process can significantly speed up drying and reduce energy consumption while maintaining phytochemicals, physical quality, and sensory attributes in final products. To overcome the downsides of conventional freeze-drying, hybrid freeze-drying methods were introduced with a great potential to provide food products at shorter drying durations, lower costs, and environmental friendliness while resulting in the same nutritive and sensory qualities as that of conventional freeze-drying in special circumstances. An overview of the most current improvements, adaptations, and applications of hybrid freeze-drying in food dehydration is given here. In this review, comparative studies are offered to characterize the drying process from the standpoint of chemical quality and sensory attributes. All the reviewed studies confirmed that the nutritional and sensory qualities of the end product can be retained using hybrid freeze-drying almost to the same extent as using single freeze-drying. It was also inferred that hybrid freeze-drying can surpass conventional freeze-drying and allow for obtaining dried products with characteristics typical of raw material if operating parameters are optimized based on product quality and energy usage.

## 1. Introduction

Freeze-drying (FD) (lyophilization) is a technique used to remove water from frozen material by sublimating ice crystals under high vacuum pressure and low temperature. In FD, the material is first frozen, after which the water is removed using sublimation (primary drying) and then using desorption (secondary drying) [1,2]. The most widespread applications of freeze-drying are in the food and pharmaceutical industries. It is also useful for a variety of products in addition to food and bulk pharmaceuticals, such as flowers, cultured microorganisms, medical equipment, and cosmetics, particularly, chemicals, pigments, and enzymes [1]. The widespread use of freeze-drying in the food industry is due to some well-known benefits over other drying methods. Food dehydration must provide good retention of vital food properties, including bioactive compounds, physical shape, and sensory quality. FD guarantees the long-term functionality of various biological materials, particularly foods with high value and those with constituents that are heat-sensitive [1,3,4]. According to the available literature, freeze-drying is an ideal method for food dehydration, allowing for long shelf-life, acceptable nutrient retention, and minimal shrinkage that creates a porous product with excellent rehydration capacity and aqueous solubility, soft texture, and reduced weight for storage [1,2,5,6]. Most degradative activities, including protein degradation, microbial action, enzymatic reaction, and non-enzymatic browning, are constrained or even prevented by the absence of liquid water and the low temperature during freeze-drying [7]. In addition, the concentration of oxygen available in the freeze-drying chamber decreases due to the high vacuum conditions, which can significantly slow down the deterioration of heat- and oxygen-sensitive components like vitamin C, anthocyanidin, and carotenoids [8]. Table 1 lists the applications, advantages, and disadvantages of freeze-drying in food.

The effects of operational factors in freeze-drying (temperature, vacuum pressure, and time) on the conservation of the key functional compounds in food-based plants (i.e., fruits, vegetables, and herbs) are extensively discussed in the literature [1,4,7]. Depending on the selected parameters, the nutritional content of the final product may increase or decrease during the freeze-drying process. In general, fruits and vegetables can be freeze-dried with higher preservation of phytochemicals and bioactivity levels when the freezing rate is faster, the drying temperature is higher, and the drying duration is shorter [4]. It is worth mentioning that ascorbic acid and anthocyanins are degraded in some food matrixes when the drying process is prolonged [4,8,9]. Further, the freezing process results in the breakdown of the cell wall and the release of bound phenolics [4,9]. Ciurzynska and Lenart [10] reported that when the freeze-drying parameters are well-defined, it can be a unique dehydration strategy that allows for the production of dried food material with chemical characteristics resembling those of the raw material.

The opportunities for effective food preservation using conventional freeze-drying are diminishing due to the sophisticated structure, activity, and metabolism of biological material [3,5,10]. As a result, there are considerable differences in the phytochemical retention in various fruits and vegetables under freeze-drying, and these variances need to be outlined [4]. Moreover, single freeze-drying has significant drawbacks, including a lengthy drying period and high energy usage (Table 1). For these reasons, the need for developing or improving the freeze-drying process for rapid drying of complex food systems is ever-increasing. Enhancements to the freeze-drying process can remarkably speed up the dehydration process and decrease energy consumption without sacrificing the product’s physicochemical and sensory qualities. To alleviate the drawbacks of conventional freeze-drying, particularly the long drying time and high energy usage, new technical solutions are being developed to upgrade the freeze-drying process and make it more time and cost-effective [1,4,11,12,13,14]. Hybrid freeze-drying as a cutting-edge technology brings together the benefits of many drying techniques and surpasses freeze-drying alone from the standpoints of product bioactivity, physical quality, sensory characteristics, operational time, energy usage, and process costs [1,2,5,6,15]. Therefore, the attention of this review is focused on recent advancements, modifications, and applications of hybrid freeze-drying in food drying. The evolution of bioactive compounds, the capacity to scavenge free radicals, and the sensory quality of plants throughout single and hybrid freeze-drying processes are all explored in this paper.

## 2. Hybrid Freeze-Drying Systems

Currently, more emphasis is being placed on the hybrid drying approach, which is a combined drying strategy that uses two or more drying techniques in continuous, simultaneous, frequent, and multi-stage modes to compensate for the drawbacks of a single drying technique [1,16,17,18,19,20]. In comparison with conventional dryers, hybrid dryers may provide various benefits, including improved product physical and sensory quality, increased retention of nutritious and bioactive components, higher energy efficiency, and better drying efficiency [21,22,23]. Recently, the hybridization of freeze-drying with additional drying methods has shown significant benefits [24,25,26,27,28]. Several types of hybrid freeze-drying technologies based on the same or comparable fundamental concepts are listed below.

## 3. Hybrid Microwave Freeze-Drying

The hybridization of microwave and freeze-drying helps to enhance the drying rate and reduce energy usage while preserving nutritional substances and creating appealing colors [29,30,31]. There are two strategies for using microwave freeze-drying: simultaneous freeze-drying and microwave drying (microwave freeze-drying), or freeze-drying followed by microwave/microwave–vacuum finish-drying (two-stage drying) [29].

In the first method, the entire dehydration process is carried out in a vacuum, and microwaves are used to supply the heat required for sublimation during freeze-drying [1,26,30]. The frozen bulk temperature rises as a result of microwave heating. As the bulk temperature increases, frozen molecules of water gain sufficient energy for a transition from the solid to the gas phase (sublimation of frozen molecules of water). These molecules move from the frozen bulk into the vacuum part of the chamber. In other words, the moisture is eliminated from the frozen region, and the food material is dried [22,26]. A schematic for a microwave freeze-drying device is shown in Figure 1.

In a pilot-scale experiment, Cao et al. [30] substituted microwave heating at intensities of 1, 1.5, and 2 W/g for contact heat to carry out the freeze-drying of barley grass. Table 2 reveals that microwave freeze-drying (MFD) retained greater flavonoids and chlorophyll contents compared with contact heat freeze-drying (FD) at the power intensities of 1 and 1.5 W/g. Due to the increased retention of chlorophyll content, MFD-dried barley grass showed a higher green color (average a* = −10.79) compared with FD-dried barley grass (average a* = −9.98). In MFD, compared with the microwave intensities of 1 and 1.5 W/g, the intensity of 2 W/g induced the destruction of flavonoids and chlorophyll owing to excessive heating. High microwave intensity quickly caused the materials to char during the last stages of MFD, lowering their quality and nutritional value. In comparison with FD, the average levels of odors were all higher in MFD (Table 3). Thus, barley grass might be dried more successfully with MFD than freeze-drying.

In a different investigation, Wang et al. [33] prepared egg yolk powder using microwave freeze-drying (MFD) with the inclusion of maltodextrin, sucrose, and trehalose carriers. To produce control powders, a single freeze-drying (FD) of egg yolk was also carried out. With the addition of carriers, MFD produced powders with better solubility (on average, about 293%) and an active antibody preservation rate (on average, about 293%) during a shorter drying period than FD. The brightness of MFD-egg yolk powder was higher than that of FD due to the lipid extrusion and water content that was still present.

In the second hybrid microwave freeze-drying method, drying is conducted in two stages: freeze pre-drying and microwave/microwave-vacuum finish-drying [29]. Studies by Fan et al. [34] and Gitter et al. [35] indicated that microwaves can be utilized to enhance the freeze-drying of vegetables as a novel dehydration technique. Microwave freeze-drying can reduce drying duration by more than half without adversely compromising the nutritional and sensory qualities of the dried food material.

Despite the unique advantages of microwave freeze-drying, ice melting and overheating might occur as a result of corona discharge and non-uniform heating [32,36]. To deal with such challenges, microwave pulse-fluidized bed freeze-drying and pulse-spouted microwave freeze-drying have been developed [11,36,37]. In microwave pulse-fluidized bed freeze-drying, microwave heating is applied to substitute for the customary conduction heating during freeze-drying. The pulsed fluidized mode may ensure the uniformity and rapidity of microwave heating, producing qualified products within a diminished drying time [12].

In a study [12], *Cordyceps militaris* was dehydrated using single freeze-drying (FD) and microwave-assisted pulse-fluidized bed freeze-drying (MPFFD), and the antioxidant and volatile chemical characteristics of the end product were evaluated. The ability to scavenge hydroxyl radicals was slightly higher in the FD-treated samples (90.4%) compared with MPFFD (85.7%). In comparison to the volatile components of fresh *Cordyceps militaris*, the relative quantities of alcohols and ketones decreased after both drying methods. Ketones accounted for 51.5% and 41.4% of the total volatile components in the FD and MPFFD samples, respectively, whereas alcohols accounted for 29.0% and 32.3%, respectively. MPFFD and FD generally produced final products with equal nutritional quality and volatile content. To obtain high-quality dried *Cordyceps militaris*, MPFFD was recommended as a powerful method for dehydration.

Pulse-spouted microwave freeze-drying was developed by incorporating pneumatic pulse agitation into microwave freeze-drying, which increased drying uniformity and sample quality [11,36]. In Jiang et al. [37], an analysis of infrared camera photos revealed that the temperature distribution within the drying sample could be improved using a lower microwave power along with a longer spouting interval and spouting period. Chinese yam cubes were subjected to pulse-spouted microwave freeze-drying (PSMFD) at microwave powers of 400, 600, and 800 W by Li et al. [36]. To avoid the extended drying time associated with processing at 400 W and the higher temperature associated with processing at 800 W, the microwave power level of 600 W was recommended as being ideal for pulse-spouted microwave freeze-drying of Chinese yam cubes. Freeze-dried yam cubes had a greater average amount of total phenols (210 mg GAE/100 g d.b.) and ascorbic acid (33 mg/100 g d.b.) than PSMFD-treated samples. The highest concentration of total phenols and ascorbic acid in PSMFD was produced with a microwave power of 600 W (160 mg GAE/100 g d.b.). It was claimed that PSMFD had better drying uniformity than microwave freeze-drying. As a result, a fluidized or spouted bed in combination with freeze-drying was regarded as an excellent method to solve the uneven problem of microwave freeze-drying [38].

To dehydrate restructured rose powder-yam chips, Hnin et al. [39] developed a microwave pulse-spouted bed freeze-drying (MPSFD) method at the cold temperature of −37 °C and a pressure of 80 Pa. The outcomes were contrasted with those obtained using microwave vacuum drying (MVD) at 50 °C and a vacuum pressure of −0.09 MPa. The anthocyanins (a group of antioxidants found in red, blue, and purple fruits and veggies) were better maintained using MPSFD (72%) as compared with MVD (50%). Electronic nose examination revealed that the odor of chips made using MPSFD was more similar to the fresh sample than that of chips made with MVD. Moreover, MPSFD-treated samples outperformed MVD-dried samples in maintaining their sensory qualities by increasing the scores for visual appearance, texture, color, and overall acceptability by 31, 16, 13, and 11%, respectively. The aforementioned research suggested that MPSFD-treated rose powder-yam chips can offer health benefits for consumers who are health-aware.

Wang et al. [40] examined the sensory quality of apple cuboids after drying using conventional freeze-drying (FD) and microwave pulse-spouted bed freeze-drying (MPSFD) methods. They used the texture profile analysis, which is the most relevant to mouthfeel sensory evaluation. The time required to produce brittle dry apple cuboids under MPSFD was determined by measuring texture at 270 and 315 min. Table 4 shows that the MPSFD samples dried for a longer period of time (315 min) were tougher than those dried for a shorter period of time (270 min). The MPSFD-produced apple cuboids had a brittle texture as a result of low cohesiveness and high hardness.

Freeze-drying assisted with microwave heating has intrinsic advantages over traditional freeze-drying. Comparing MFD products to FD-dried food, MFD products exhibit higher volatile retention levels. MFD can produce dried food with nearly the same quality as freeze-drying and even more, but there are still a lot of practical issues that need to be overcome. Vacuum pressure and microwave power in hybrid microwave freeze-drying systems are key factors in determining product quality [30,36]. High microwave power causes freeze-drying to fail if the sample temperature becomes substantially greater than the triple point (solid, liquid, and gas). Significant losses in total phenols and ascorbic acid as well as poor color and texture are among the negative consequences caused by overheating during MFD. On the other hand, drying at insufficient microwave power may also result in greater losses of total phenols and ascorbic acid due to the extended drying period. Therefore, regulating the process parameters in MFD is highly required to produce food products with superior nutritive and sensory quality. Corona discharge and non-uniform heating were the two main technical issues that were found to be obstacles to the viability of MFD. To overcome these challenges, microwave-assisted pulse-fluidized bed freeze-drying and microwave pulse-spouted bed freeze-drying are good alternatives for MFD.

## 4. Hybrid Infrared Freeze-Drying

Infrared freeze-drying is one of the most promising drying strategies among the combined freeze-drying-based approaches, which yields better product quality while providing a faster drying process [16,17,23,25]. Infrared freeze-drying is an innovative drying process that uses infrared rather than electric heating plates to generate energy for ice sublimation, which increases heat transfer rates [3,23]. Infrared radiation is a type of electromagnetic wave with a wavelength within the range of 0.78 to 1000 µm. Uniform heating and high energy transfer efficiency are two advantages of infrared (IR) heating, which is mostly attributable to the fact that infrared radiation enters a product to a specific depth and raises its temperature without heating the air around it. A medium is not necessary for the infrared-based heating process [3]. A schematic view of an infrared freeze-dryer is shown in Figure 2.

Khampakool et al. [16] used four different freeze-drying techniques for banana snacks: single freeze-drying (FD), continuous near-infrared freeze-drying (IRFD) at a power density of 2.7 kW/m^2^, IRFD-2.7 kW/m^2^ at 20% weight reduction, and IRFD-2.7 kW/m^2^ at 20% weight reduction to 4.0 kW/m^2^ at 0 °C. Although a chemical analysis was not conducted, IRFD could improve the sensory quality of banana chips compared with FD, as listed in Table 5. This was attributed to more rapid drying during IRFD trials than during FD. This study showed how IRFD can be used to produce fruit chips with enhanced sensory values in particular crispness.

Wu et al. [3] used hybrid infrared freeze-drying (IRFD) and single freeze-drying (FD) techniques to dry *Cordyceps militaris* at temperatures of 40, 50, 60, and 70 °C. They assessed the antioxidant activities, nutritional characteristics, and main volatile compounds in the final products. IRFD resulted in slightly higher hydroxyl radical scavenging activities (~93 and 92%, respectively) than FD at the drying temperatures of 50 and 60 °C, although the difference was not statistically significant (*p* > 0.05). The antioxidant activity, total phenolics, and reducing power in both drying techniques were higher at 50 and 60 °C than that at 40 and 70 °C. This finding revealed that high drying temperatures as well as lengthy drying times did not favor the retention of bioactivity in *Cordyceps militaris*. Comparing the retention of the volatile compounds in IRFD- and FD-treated samples at a constant drying temperature, the former showed greater retention of volatiles, including 3-octanone, 1-octen-3-ol, 3-octanol, and 1,3-octadiene. This showed that aroma retention was made easier with quick drying times. Based on the findings of Wu et al. [3], IRFD could be a potential strategy for producing high-quality dried *Cordyceps militaris*.

Ee et al. [2] investigated the impact of two-stage infrared freeze-drying (FDIR) and freeze-drying alone (FD) on the nutritive and sensory qualities of kedondong fruits (*Spondias dulcis*). The hybridization process was divided into two stages: freeze-drying (FD period: 6 or 12 h), and then infrared heating at 60 °C. When freeze-drying in the hybrid approach was extended to 12 h (FD12IR), the polyphenol content (5.46 mg GAE/g d.b.) was statistically identical to the polyphenol content obtained using FD (5.56 mg GAE/g d.b.) (*p* > 0.05). The same result was observed for the ABTS antioxidant activity of FD (9.67 μM Trolox equivalent/100 g d.b.) and FD12IR (9.55 μM Trolox equivalent/100 g d.b.) samples. However, the DDPH antioxidant activity in FD-treated kedondong fruits was 44.73% more than that in the FD12IR-treated samples. The polyphenol content and antioxidant properties (ABTS and DPPH) preserved in FD12IR were about 45, 34, and 81%, respectively, more than that in FD6IR. The just about right (JAR) scores of kedondong notes and kedondong aromas in FD and FD12IR were 54–56 and 54–50, respectively. FD12IR was found able to improve the retention of bioactivity, notes, and aromas in kedondong fruits.

Chives were dried using catalytic infrared drying at 70 °C prior to freeze-drying at −80 °C (CIRD+FD) and single freeze-drying (FD) [17]. FD had significantly higher vitamin C (61.12%), allicin (28.25%), and chlorophyll (80.56%) retention (*p* < 0.05) than CIRD+FD samples with an average retention of 52.21, 75.89, and 22.56%, respectively. The aroma of the FD- and CIRD+FD-treated samples did not significantly (*p* > 0.05) differ at the intermediate moisture percentages of 80–85%. The use of CIRD in conjunction with traditional freeze-drying could generally offer a way for preparing high-quality dried chives, according to a thorough analysis of the product quality and volatile components.

In another study by Chao et al. [41], freeze pre-drying (at the cold trap temperature of −50 °C and vacuum pressure 10 Pa for 15, 25, and 35 h) and far-infrared finish-drying at the temperature of 60 °C for 1, 2, and 4 h (FD+FIRD) were combined to dry seed-used pumpkin. Single freeze-drying was also used for effective comparison. The pumpkin slices were dried to reach a final moisture content of less than 7% (w.b.). Generally, the free phenolic content of the FD+FIRD samples was 14.97–26.60% higher than that of the FD depending on FIRD duration. With reference to FD, a higher concentration of chlorogenic acid (a phenolic compound widely found in fruits and vegetables) was found in FD35+FIRD1 (206.13 3.61 g/g d.b.), which was related to a decrease in the oxygen-induced oxidation process during FD. FD25+FIRD1 preserved the content of p-coumaric acid (a potent phenolic compound) by 32.23% in comparison to FD. The retention of total carotenoids and β-carotene was slightly higher in pure freeze-drying compared with FD+FIRD.

Hnin et al. [25] dried rose-flavored yogurt melts by combining infrared freeze-drying and an infrared dryer (IRFD+IRD). The samples were dried for a predetermined time (2, 2.5, or 3 h) using infrared freeze-drying (IRFD) before being transferred to an infrared dryer and dried to a desired moisture content of 5% (w.b.). The total phenolic contents in the products from all IRFD and IRD combinations (average 3.80 mg GAE/100 g d.b.) was lower than in the IRFD sample (4.3 mg GAE/100 g d.b.). This may have been due to the higher drying temperature during the second step of IRD, which led to phenolic compounds being lost. The order of the content of total phenolics was IRFD3+IRD ~ IRFD (*p* > 0.05) > IRFD2.5+IRD > IRFD2+IRD. In this investigation, the IRFD samples had a higher content of total anthocyanins (0.54 mg/g d.b.) than the IRFD3+IRD (0.51 mg/g d.b.), IRFD2.5+IRD (0.43 mg/g d.b.) and IRFD+IRD (0.39 mg/g d.b.) samples. The hydroxyl radical scavenging activity of the samples prepared using IRFD was the highest (89%), whereas IRFD2+IRD resulted in the lowest scavenging activity (77%). For the preparation of yogurt melts with rose flavor, the IRFD3+IRD technique is a promising method in terms of nutritive quality.

In Lao et al. [23], kale yogurt melts were subjected to infrared freeze-drying for a predefined time (3, 4, and 5 h) prior to microwave vacuum drying (IRFD-MVD) until the moisture content was less than 5% (d.b.). According to Figure 3, the highest preservation of chlorophylls was found in the samples prepared using FD followed by IRFD and IRFD5-MVD. The preservation of carotenoids followed the same trend. The total phenolic content in the IRFD+MVD samples in all cases was almost the same as in the IRFD sample (*p* > 0.05) but was lower than the FD-treated sample. The antioxidant activity rank of the samples was: FD > IRFD > IRFD5-MVD > IRFD4-MVD > IRFD3-MVD. It was concluded that for materials that are challenging to shape at normal drying temperatures and pressures, IRFD and IRFD-MVD under suitable conditions (4 and 5 h) may be an alternative drying process to produce high-quality end products. The highest-scoring sample for the sensory quality was the IRFD sample (Table 6).

In a study by Antal et al. [5], mid-infrared at temperatures of 40, 50, and 60 °C (each for 5 min) was incorporated into freeze-drying (MIR-FD) and used for pear dehydration. Freeze-drying alone (FD) at the temperature of −24 °C and absolute pressure of 85–95 Pa was also applied for the effective evaluation of hybrid drying performance. MIR-FD had superiority over FD in terms of preserving a higher content of total phenolics and antioxidant activity by 16.53 and 93.63%, respectively. The MIR-FD technique produced dried pear cubes with a higher rehydration ratio (3.37) because of lower physical changes, owing to uniform heating and lower drying than FD.

It has been demonstrated that IR-assisted freeze-drying yields products with nutritional and sensory qualities that are comparable to single freeze-dried samples. A thick structure on the product surface layer and large pores in the material center are usually seen during various infrared freeze-drying trails. Collapse happens as a result of an excessive rise in product temperature when exposed to IRFD, which has an adverse impact on the physicochemical properties of the product [22,41,42]. To avoid collapse or boiling during infrared freeze-drying, the duration of infrared radiation should be adequately managed. Infrared freeze-drying assisted with microwave vacuum drying also can be used as an alternative dehydration strategy for materials that are challenging to dry at normal drying pressures and temperatures. Under optimal conditions, infrared microwave vacuum freeze-drying has the potential for large-scale industrial production due to the faster drying time and capacity to create high-quality dried food products [23].

## 5. Hybrid Hot Air Freeze-Drying

Hybrid hot air freeze-drying of food has been reported to significantly reduce drying duration and energy usage compared with pure freeze-drying [2,38]. The average rate of mass transfer in hybrid hot air freeze-drying is higher, resulting in a lower drying time than in pure freeze-drying. Free water is quickly eliminated at the onset of the hot-air-drying process, causing an intense mass transfer and faster freeze-drying rate [2,43].

Alhamid et al. [44] evaluated the influence of inserting hot air from a reservoir into vacuum freeze-drying. Although vacuum freeze-drying with a hot air reservoir could save drying time, the moisture content of the product was higher than it was during vacuum freeze-drying. Vacuum freeze-drying with hot air insertion requires additional vacuum pump capability, especially in flow rate and ultimate vacuum, to maintain the chamber pressure below the triple point condition.

To ensure product quality, Zhang et al. [24] used hot air as an alternative to the desorption process of freeze-drying [24]. They evaluated the combination of hot air drying and freeze-vacuum drying (FVD+AD) for drying kiwifruit. Freeze-vacuum drying alone (FVD) was also carried out to compare the results. For FVD+AD, the values of bioactive components, including ascorbic acid (183.99 mg/100 g, d.b.), total flavonoids (0.84 mg _rutin_/g, d.b.), and DPPH antioxidant activity (306.10%, d.b.) were lower than that of freeze vacuum-drying (FVD), in which the ascorbic acid, total flavonoids, and DPPH radical scavenging activity were 234.75 mg/100 g, d.b., 1.56 mg _rutin_/g, d.b., and 368.68%, d.b., respectively. The FVD+AD sample had a hardness that was around 32% higher than the FVD sample. More color retention (approximately 56%) and the presence of volatile components resembling those in fresh kiwifruit allowed FVD+AD to surpass FVD.

Ee et al. [2] investigated how the bioactivity of kedondong fruit (*Spondias dulcis*) was affected by a two-stage drying procedure that combined hot air and freeze-drying (FD+AD). First, the material was frozen for 6 or 12 h, and then it underwent a second stage of hot air drying at a temperature of 60 °C. The FD-treated sample had higher levels of total phenol (5.56 mg GAE/g d.b.), ABTS antioxidant activity (9.67 M Trolox equivalent/100 g d.b.), and DPPH radical scavenging activity (216.41 mg AA/100 g d.b.) than the FD+AD product in both freeze-drying durations. Due to an increase in the preservation of total phenol content as well as ABTS and DPPH antioxidant properties of about 26, 22, and 30%, respectively, FD12+AD outperformed FD6+AD.

To dry button mushroom slices, Pei et al. [45] coupled freeze-drying (FD) with three different dehydration techniques: freeze-drying with hot air drying (FD+AD), freeze-drying with vacuum drying (FD+VD), and freeze-drying with microwave vacuum drying (FD+MVD). The nutrient retention, color, and texture of products were all thoroughly examined. Table 7 shows that with the exception of vitamin C content, the FD, FD+MVD, and FD+VD approaches considerably (*p* < 0.05) outperformed FD+AD in terms of maintaining protein, reducing sugar, and total sugar contents. FD+AD resulted in dried mushrooms with significantly lower bioactivity than the other drying techniques, which may be related to the Maillard reaction under the AD condition. The hardness of the FD+VD and FD+MVD products did not significantly differ (*p* > 0.05). However, the hardness of FD+AD products (230.43 g) was significantly higher (*p* < 0.05) than the others. For the same moisture content changing points, FD+VD and FD+MVD products had closer colors to FD in comparison with the FD+AD products. This may have been attributed to the presence of vacuum conditions in the FD, FD+MVD, and FD+VD approaches. As a high-quality and effective drying technique, FD+MVD can replace FD methods to manufacture high-quality dehydrated products.

A combination of hot air and freeze-drying techniques is useful because they have a synergistic effect when drying foods with high moisture content. Dehydration efficiency successfully increases using hybrid hot air-freeze drying. Future research should assess different food items to show the effectiveness of hot air freeze-drying on the nutritive quality of plant-based material.

## 6. Hybrid Explosion Puffing Freeze-Drying

The fundamental method in the explosive puffing drying system (also referred to as instant controlled pressure drop texturing) is to first loosen the product texture at high pressures (typically 0.4 MPa) and high temperatures (roughly 100 °C) and then quickly and thoroughly remove moisture from the product under atmospheric pressure (under vacuum) to create a porous structure [1,46,47]. This technique also relies on thermo-mechanical processes linked to samples that are quickly exposed to saturated steam (between 0.1 and 0.6 MPa), followed by a sudden pressure drop into the vacuum.

Explosive puffing drying is not recommended for the drying of raw agricultural products with high moisture content including fresh fruits and vegetables [1,21,46,47]. Typically, the moisture content of the food is decreased to the appropriate level (around 30% (w.b.)) prior to explosive puffing drying. Many dehydration techniques, such as hot air, microwave, and freeze-drying, have been introduced prior to explosive puffing drying to diminish the moisture content of raw materials [21,47,48]. Regarding energy usage and product quality, the creative combination of freeze-drying and explosive puffing drying with optimized processing conditions could be a viable solution for dry foods, particularly restructured fruit and vegetable chips [1,47].

The hybrid explosion puffing freeze-drying method has been applied successfully to jackfruit bulb chips [49], black mulberries [21], papaya chips [46], carrot–potato chips [47], starch-free snacks [50], and pumpkin chips [48]. In a comparative study [48], pumpkin slices were dried using single freeze-drying (FD) at the vacuum pressures of 0.1, 0.15, and 0.2 mbar and the combination of freeze-drying and explosive puffing drying at two stages (FD+EPD). The EPD at the second stage of hybrid drying was conducted at a temperature of 90 °C and pressure of 190 kPa. The average retention of DPPH inhibition, total phenolic content, and total carotenoids (yellow, orange, and red pigments) in single FD were 78.94%, 1381.11 mg/100 g d.b., and 19.04 mg/100 g d.b, respectively, which were about 3, 4.5, and 5%, respectively, more than that in FD+EFD. However, hybrid drying was more successful in improving the average bulk density (6.82%), volumetric expansion (27.78%), and rehydration ratio (20.08%) of pumpkin chips compared with FD. Hence, the FD-EPD method is a potent drying technique that could be used to turn pumpkin slices into snack foods like chips while maintaining their functional and textural properties with less energy.

Köprüalan et al. [27] evaluated the effects of freeze-drying combined with explosion puffing drying (FD+EFD) and microwave freeze-drying as a pre-drying method before explosion puffing drying (MD+FD+EPD) on the chemical quality and sensory attributes of white cheese snacks. The integration of MD and FD into MD+FD+EPD techniques resulted in higher protein content (60.72 g/100 g d.b.) and improved sensory properties due to lower salt 5.57 (g/100 g d.b.) and higher fat (22.85 g/100 g d.b.) contents in comparison with FD+EPD, in which the values of protein, salt, and fat were 57.52, 9.81, and 19.69 g/100 g d.b., respectively. Regarding the FD+EPD chips, the improved porous structure of the semi-dried materials may be more favorable to additional volume expansion during explosion puffing drying. This is likely because it would provide more capillary paths for releasing internal water vapor, resulting in a crispier texture and more uniform porosity.

Explosion puffing freeze-drying is a viable alternative to freeze-drying alone, according to the aforementioned studies [1,47]. This is a result of the production of merchandise that is comparable to freeze-dried foods in terms of nutrition and flavor.

## 7. Hybrid Electro-Hydrodynamic Freeze-Drying

In comparison with convective freeze-drying, electro-hydrodynamic drying needs a less complex system with lower energy demand. To improve the drying rate, this method uses a high electric field composed of one or more point electrodes and a plate electrode [38]. In order to produce ionized air-constituent forms, an electrode with strong electric fields is used. An ionic wind is produced when air ions travel in a strong electric field. This ionic wind propels molecules of water vapor into the electric field, facilitating their removal and accelerating drying [51]. In particular, heat-sensitive foods and biological products benefit significantly from electro-hydrodynamic drying, which is a new non-thermal and energy-saving drying technology that has advantages over conventional drying techniques and has quickly become a research hotspot [28].

Although the quality of products dried using electro-hydrodynamics is inferior to that obtained using freeze-drying, combining the two drying methods reduces the drawbacks of a single drying technique and allows the advantages of both methods including product high quality and low cost [38,52]. Studies on the application of electro-hydrodynamic freeze-drying are limited and include only reports on shrimp [52] and sea cucumber [53].

Bai et al. [53] studied the effect of freeze-drying and electro-hydrodynamic freeze-drying (EHFD) on sea cucumbers. Samples dried using FD had higher protein content (~50 g/100 g d.b.) than the samples dried using EHFD (~43 g/100 g d.b.). Further, the sensory quality of the FD-obtained samples achieved a score of 9.5, which was better compared with the EHFD score of 9.

In another study, Hu et al. [52] used electro-hydrodynamic freeze-drying (EHFD) and freeze-drying for dehydration of shrimp. Hybrid drying had a drying rate that was 32.56% higher than FD, although FD performed better than EHFD in terms of shrinkage, rehydration rate, and sensory attributes. Samples subjected to FD received a sensory quality score of 9.2, but those subjected to EHFD received a score of 8.5. The final shrinkage values for FD and EHFD drying were, respectively, 7.32% and 17.35%. This shows that EHFD significantly reduces shrinking.

When electro-hydrodynamic drying and freeze-drying are combined, better product quality and higher drying efficiency may be obtained.

## 8. Applications of Hybrid Freeze-Drying Techniques in Food Dehydration

Hybrid freeze-drying is a promising technology that is frequently utilized in the drying of food, particularly fruits and vegetables; however; there are few reports on its application in the drying of meat and poultry, seafood, and beverages [1,36]. Recently, hybrid freeze-drying was applied in numerous applications to process fresh products (Table 8). Khampakool et al. [6] investigated the potential of freeze-drying alone and infrared-assisted freeze-drying (IRFD) for the dehydration of edible insects (*Proteotia brevitarsis* larva). The sublimation energy required for fast freeze-drying was provided with an infrared lamp. When compared with FD, continuous IRFD considerably (*p* < 0.05) reduced drying time and energy usage by 91 and 90%, respectively. With respect to FD, IRFD products displayed considerably decreased chewiness, hardness, and higher protein content (*p* < 0.05). Proline (3.84–5.54 g/100 g) and glutamic acid (6.30–7.29 g/100 g) were both better conserved using IRFD.

Hybrid freeze-drying can also be used for purposes other than food drying. For example, the applicability and feasibility of microwave-assisted freeze-drying for various monoclonal antibody formulations, as well as its impact on the attributes of the final product, were assessed. It was confirmed that microwave-assisted freeze-drying remarkably reduces drying durations while retaining product quality, which is also a prerequisite for continuous processing. In fact, the concept of continuous pharmaceutical freeze-drying with microwave assistance appears interesting [54]. Gitter et al. [54] also investigated four distinct formulations of monoclonal antibodies using various glass-forming excipients: trehalose, sucrose, and arginine phosphate. These formulations were dehydrated using freeze-drying alone and microwave-assisted freeze-drying (MFD), stored for 24 weeks, and then tested for solid-state and protein-related quality characteristics. MFD significantly shortened drying time by over 75% compared with FD. As a result, MFD provided a potent and distinctive option for the rapid supply of freeze-dried biological products while maintaining product quality.

**Table 8 foods-12-03437-t008:** Some recent research on the hybrid freeze-drying of food products.

Food Category		Hybrid Freeze-Drying Type	Key Highlights	References
Fruits and Vegetables	Mango, pitaya, and papaya chips	Integrated freeze-drying and explosion puffing drying (FD+EPD)	The content of AA, TC, TP, and TF in FD samples was relatively more than that in FD+EPD; however, the TSS and TA contents were almost the same (*p* > 0.05) in both methods. Hybrid-dried samples had better color retention and sensory scores compared with FD.	[55]
	Kiwifruits	Combined hot air drying and freeze vacuum-drying (FVD+AD)	FD outperformed FVD+AD in the contents of TF, AA, and AC by 21.61, 24.26, and 16.97%, respectively. FVD+AD improved microstructure and retained volatile compounds and color compared with FD.	[24]
	Button mushrooms	Combined microwave freeze-drying (FD+MVD)	Protein, vitamin c, reducing sugar, and total sugar in the FD+MVD samples were about 4, 42, 13, and 15%, respectively, more than in the FD samples.	[45]
	Chinese yam cubes	Pulse-spouted microwave freeze-drying (PSMFD)	Total phenolics and ascorbic acid were better preserved using single FD, both by about 33% compared with PSMFD. The TP and AA of the PSMFD samples were highest at 600 W (162.25 mg GAE/100 g d.b. and 29 mg/100 g d.b., respectively) compared with 400 and 800 W.	[36]
	Pumpkin chips	Combined freeze-drying and explosive puffing drying (FD+EPD)	The average retention of DPPH inhibition, TP, and TC in single FD were about 3, 4.5, and 5%, respectively, more than that in FD+EFD.	[48]
	Okra snacks	Combined microwave vacuum drying and freeze-drying (FD+MVD)	TF and TP in the FD sample were both about 5% higher than in the FD-MVD sample. The FD+MVD sample exhibited higher IC50 and ABTS antioxidant properties by about 8 and 6%, respectively.	[56]
	Pears	Mid-infrared-freeze drying (MIR+FD)	MIR+FD had higher rehydration capability and produced products with lower hardness and larger quantities of TP (1.37-fold) and AC (1.94-fold) with respect to FD.	[5]
	Chives	Catalytic infrared drying prior to freeze-drying (CIRD+FD)	FD samples had significantly higher vitamin C (61.12%), allicin (28.25%), and chlorophyll (80.56%) retention (*p* < 0.05) than CIRD-FD samples with an average retention of 52.21, 75.89, and 22.56%, respectively.	[17]
	Jackfruit (*Artocarpus heterophyllus* L.)	Instant controlled pressure drop-assisted hot air drying (AD+DIC)	The quantities of TP (1.33-fold), TC (2.13-fold), DPPH (1.41-fold), FRAP (1.12-fold), and ABTS (1.23-fold) antioxidant activity were larger in FD. FD+DIC chips had a smaller color change (6.5) and modified texture (crispness 19, hardness 42 N) compared with FD.	[49]
Herb	*Flos Sophorae Immaturus*	Ultrasound-assisted freeze-drying (UFD)	UFD improved the retention of TF compared with FD.	[57]
Meat and Poultry	Duck egg white protein powders	Combined two-stage freeze-drying and microwave–vacuum (FD+MVD)	Higher FC (1.26-fold), ESI (1.73-fold), and color retention (2.75-fold), hardness (3-fold) and lower EAI and FS were found in FD products.	[58]
Seafood	Sea cucumber	Electro-hydrodynamic freeze-drying (EHFD)	FD-dried samples displayed higher protein content (14%) along with better sensory qualities.	[53]
	Fish muscle	Microwave freeze-drying (MFD)	MFD reduced the concentration of organochlorine pesticides extracted from fish muscle tissue.	[59]
	Sea cucumber (*Stichopus japonicus*)	Microwave freeze-drying (MFD)	The FD- and MFD-treated samples had the content of amino acids in dry matter (~73 g/100 g) and sensory values (~9.5) (*p* < 0.05).	[32]

TSS: total soluble solid; TA: titratable acidity; AA: ascorbic acid; TP: total phenolics; TC: total carotenoids; TF: total flavonoids, AC: antioxidant capacity; FS: foaming stability; FC: foaming capacity; EAI: emulsifying activities; ESI: emulsifying stabilities.

## 9. Benefits of a Hybrid Freeze-Drying System in Food Drying

Convectional freeze dryers are reported to have 10% efficiency due to a prolonged freezing and drying duration, which consumes more energy [1,3]. The breakdown of the energy costs for a typical freeze-drying process was provided by Ratti [60]. Around 45% of the total energy cost is associated with the sublimation process, with the next largest contributions coming from vacuum (26%), condensation (25%), and freezing (4%). High energy is required during single freeze-drying to maintain condenser systems running at very low temperatures and provide a low water vapor pressure environment. In other words, it is crucial to pay close attention to energy needs throughout the sublimation stage.

Most hybridization of freeze-drying is designed to improve the energy efficiency of existing conventional freeze-drying techniques [1,23,61]. Hybrid freeze dryers can enhance drying kinetics, reduce drying time and energy consumption, and improve product physical and sensory characteristics while maintaining nutritive quality similar to pure freeze dryers [14,23]. For instance, in a study on the drying of okra snacks, it was found that single freeze-drying had a specific energy consumption of about 75 kWh/kg_H2O_ as opposed to hot air drying, in which the specific energy consumption was about 50 kWh/kg_H2O_. The specific energy consumption was reduced to 20 kWh/kg_H2O_ by combining freeze-drying and microwave vacuum-drying [56]. In another study [32], sea cucumber was dried using microwave freeze-drying, and it was discovered that MFD reduced drying time and energy consumption by 40 and 32%, respectively. The use of ultrasound-assisted freeze-drying by Carrion et al. [62] resulted in a 74% reduction in the drying time for button mushrooms. Antal [42] reported that infrared freeze-drying (IR+FD) and hot air-assisted freeze-drying (AD+FD) outperformed conventional freeze-drying with a 45.5 and 27.3%, respectively, reduction in drying time.

Based on our review of the literature, numerous individual FD-based hybrid drying technologies have consistently resulted in products with quality similar to single freeze-drying and, in some cases, better than that of conventional freeze-dried products. However, the cost of drying equipment is also a crucial factor that must be taken into account in addition to achieving higher product quality. The capital, operational, and maintenance costs of freeze-drying units are four to eight times higher than conventional drying units such as hot air drying [1]. According to reports, FD-based hybrid drying technologies outperform conventional FD in terms of drying effectiveness, energy usage, and equipment operation costs. It can also guarantee the preservation of nutritive quality and the improvement of sensory attributes. It is also inferred that hybrid freeze-drying can surpass conventional freeze-drying and allow for obtaining dried products with characteristics typical of raw material if operating parameters are optimized based on product quality and energy usage.

The proper combination of heating methods, combined with the synergistic impact of hybrid freeze-drying, may improve product quality by limiting microstructural changes and maintaining color, nutrients, bioactive chemicals, and volatile components [3].

## 10. Scaling up Hybrid Freeze-Drying System to the Industry Level

Hybrid freeze-drying technology has demonstrated a number of superiorities, including higher drying and energy efficiency, effective moisture elimination, and improved product bioactivity as well as sensory quality [1,6,12,17]. This is possible if in-depth research is performed and if hybrid freeze-drying is carried out under optimal operating conditions [14,25]. To further comprehend the association between hybrid freeze-drying and cost performance aspects, more thorough techno-economic studies may be required if such technology is implemented [1,14]. The manufacturing costs were examined in [63] for a typical freeze-dried unit dose produced with both a pilot-scale and an industrial-scale freeze-drier. According to Table 9, while the capital costs increased by a factor of 20, the operational cost per cycle carried out in the industrial freeze-dryer increased by a factor of 30 in comparison with the pilot-scale freeze-dryer. This discrepancy is expected given that machine prices often increase less than linearly with machine dimensions.

It is fascinating to compare the expenses per dose for the two arrangements, even in light of this significant increase in price. Despite the fact that an industrial freeze-dryer has a cycle cost that is 25 times higher overall, the cost per dosage is six times lower due to the rise in productivity. Indeed, this is something that should be anticipated given that a main goal of scaling up a process is to lower costs and boost productivity.

In terms of dried product quality, drying effectiveness, and equipment costs, the new FD-based hybrid drying technologies have not yet been directly compared. This may be due to the fact that these drying methods are still in the research and development stage and have not been extensively applied to industrial production. Furthermore, the majority of hybrid freeze dryers are established on a laboratory scale, and scaling up hybrid freeze dryers seems more challenging than traditional freeze dryers. As a result, developing a “protocol” or “guideline” for transitioning from the laboratory to pilot-scale hybrid freeze-drying is beneficial [1,14]. In order to assess the combined effects of energy, the environment, and cost, hybrid freeze-drying systems should also go through a rigorous full-scale life cycle analysis and a life cycle cost analysis. This would help to support the usefulness of hybrid freeze-drying [14]. Most hybrid freeze dryers are designed and operated in batch mode, while the continuous mode has not been extensively investigated [1]. As a result, it is worthwhile to conduct additional research in these areas to encompass a broader variety of design and operating circumstances as well as a more diverse spectrum of food and bio-products [14].

## 11. Conclusions

Freeze-drying is a well-accepted drying technology in which product morphological, biochemical, and immunological properties can be preserved at low processing temperatures. Currently, the applications of this technology range from easy food preservation to sophisticated biotechnological or medicinal items and proliferating microbes and fungi. Despite wide applicability and unmatched advantages, freeze-drying has been considered as the most expensive dehydration process due to longer drying duration, excessive energy consumption, and high capital, operational, and maintenance costs. Therefore, FD advancements can greatly increase drying rates and reduce energy consumption without sacrificing product quality. A wise hybridization of freeze-drying and other drying techniques can be beneficial in overcoming the drawbacks of FD. The basic approach behind hybrid freeze-drying is to increase energy efficiency while maintaining high product quality. Many researchers have demonstrated that the incorporation of other drying methods, such as microwave drying, infrared drying, hot air drying, explosion puffing drying, ultrasound drying, and electro-hydrodynamic drying, into FD can assist in achieving shorter drying times, lower energy requirements, and improved nutritive and sensory quality. From the viewpoint of biochemical and sensory qualities, a precise choice has to be made in selecting technologies and in the various process and product parameters during hybrid freeze-drying to focus on the synergistic benefits of different approaches and to minimize the limitations of individual techniques. If the operating parameters of hybrid freeze-drying systems are well-defined, the nutritional status of products matches that of freeze-dried food material. Consequently, the results of the reviewed studies suggest that hybrid freeze-drying can be a promising approach for achieving a time-saving, energy-efficient, and cost-effective dehydration process that produces high-quality dried products.

## Figures and Tables

**Figure 1 foods-12-03437-f001:**
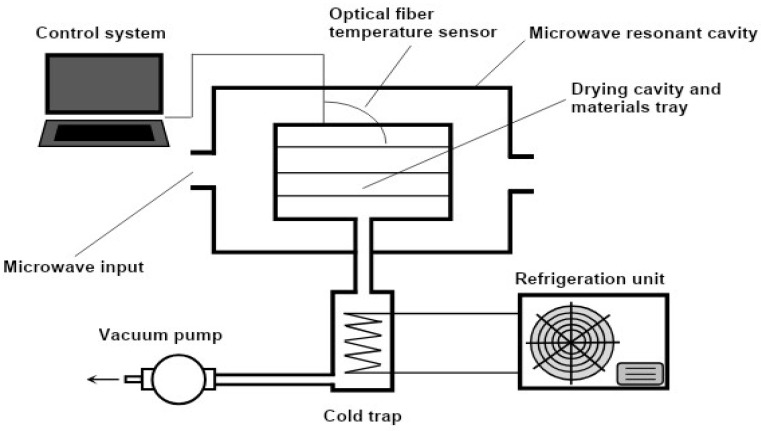
Scheme diagram showing hybrid microwave freeze-drying [32].

**Figure 2 foods-12-03437-f002:**
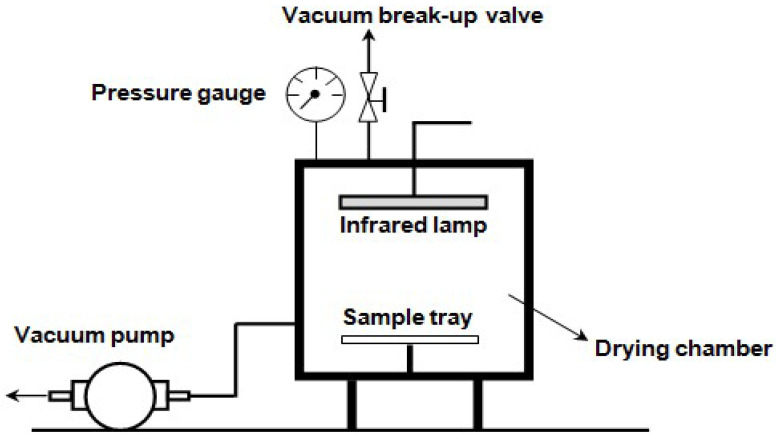
Schematic diagram showing a hybrid infrared freeze-drying [41].

**Figure 3 foods-12-03437-f003:**
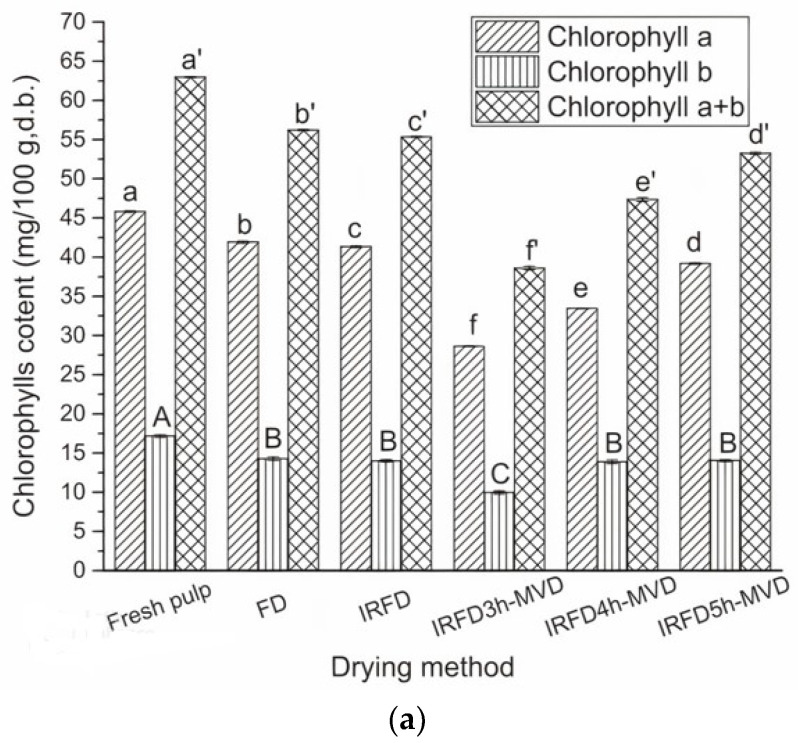
(**a**) Chlorophyll content, (**b**) carotenoid content, (**c**) total phenolic content, and (**d**) ABTS radical scavenging capacities of samples prepared using freeze drying (FD) and different infrared and microwave-vacuum assisted freeze-drying (IRFD) trials (different letters within the same column indicate significant differences (*p* < 0.05)).

**Table 1 foods-12-03437-t001:** Applications, advantages, and disadvantages of freeze-drying in food [1,2,5,6,8].

Applications in Food	Advantages	Disadvantages
Freeze-drying is useful when the product meets one or more of the following criteria: − It is unstable. − It is heat stable. − Quick and complete rehydration is required. − The product is of high value. − Weight must be minimized. − Frozen or chilled storage is not appropriate. − Meat, vegetables, fruits, medical/culinary herbs, fungus, and micro-powders are among the categories of freeze-dried foods.	Creating porous friable structures such as in snacks.	Not all foods can be freeze-dried.
	Retention of morphological, biochemical, and immunological properties.	Slow process, in particular, in the case of large objects.
	Maintaining the sensory quality (flavors, smells, and color).	High energy consumption.
	Producing lightweight food and convenient storage, transport, and distribution.	High capital, operational, and maintenance costs.
	Sample stability at room temperature (no need for refrigeration).	No space savings due to retaining the food’s cellular structure.
	Long-term storage.	Airtight containers are required for long-term storage.
	Minimum damage to the heat-labile material.	Some dislike the dry and styrofoam texture.
	Allowing very quick rehydration to restore the food’s organoleptic properties and weight.	Freeze-drying is facing difficult challenges as the sensitivity, complexity, and price of treated products steadily rise.
	Reducing water content and inhibiting the action of microorganisms and enzymes.	Freezing damage can occur with labile products, such as liposomes, proteins, and viruses.

**Table 2 foods-12-03437-t002:** Flavonoid and chlorophyll content as well as greenness value (a*) of FD- and MFD-dried barley grass at different microwave intensities [30].

Method	Power Intensity (W/g)	Flavonoid Content (g/kg, d.b.)	Chlorophyll Content (g/kg, d.b.)	a*
FD	2	11.78 ± 0.11 ^d^	13.75 ± 0.22 ^e^	−11.77 ± 0.01 ^a^
	1.5	11.29 ± 0.42 ^d^	11.41 ± 0.31 ^c^	−11.46 ± 0.04 ^b^
	1	10.12 ± 0.21 ^c^	10.15 ± 0.10 ^b^	−6.72 ± 0.04 ^e^
MFD	2	9.52 ± 0.13 ^b^	10.42 ± 0.30 ^b^	−9.56 ± 0.06 ^d^
	1.5	11.65 ± 0.22 ^d^	12.75 ± 0.31 ^d^	−11.46 ± 0.04 ^b^
	1	11.45 ± 0.21 ^d^	12.62 ± 0.20 ^d^	−11.34 ± 0.03 ^b^

Mean value ± standard error of three determinations; different letters mean a difference at *p* ≤ 0.05.

**Table 3 foods-12-03437-t003:** Main odors of FD- and MFD-dried barley grass at different microwave intensities [30].

Method	Power Intensity (W/g)	S1	S2	S3	S5	S8
FD	2	1.55 ± 0.02 ^ab^	1.49 ± 0.02 ^ab^	1.35 ± 0.03 ^a^	1.64 ± 0.02 ^ab^	1.75 ± 0.01 ^bc^
	1.5	1.51 ± 0.01 ^a^	1.50 ± 0.02 ^ab^	1.32 ± 0.03 ^a^	1.64 ± 0.02 ^ab^	1.70 ± 0.02 ^ab^
	1	1.45 ± 0.01 ^a^	1.45 ± 0.02 ^a^	1.31 ± 0.01 ^a^	1.57 ± 0.03 ^a^	1.65 ± 0.08 ^a^
MFD	2	1.74 ± 0.04 ^c^	1.59 ± 0.07 ^b^	1.28 ± 0.02 ^a^	1.72 ± 0.08 ^ab^	1.89 ± 0.04 ^e^
	1.5	1.68 ± 0.04 ^c^	1.57 ± 0.02 ^ab^	1.33 ± 0.01 ^a^	1.75 ± 0.02 ^b^	1.80 ± 0.02 ^cd^
	1	1.65 ± 0.04 ^bc^	1.56 ± 0.04 ^ab^	1.26 ± 0.02 ^a^	1.75 ± 0.05 ^b^	1.82 ± 0.02 ^d^

Mean value ± standard error of three determinations; different letters mean a difference at *p* ≤ 0.05. S1: aromatic, S2: amine, S3: sulfides, S5: esters, S8: ammonia.

**Table 4 foods-12-03437-t004:** Texture of apple cuboids at the final stage of MPSFD compared with FD [40].

Drying	Hardness	Adhesiveness	Springiness	Cohesiveness	Chewiness	Resilience
MPSFD-270	351.06 ± 72.19 ^b^	−0.29 ± 0.09 ^a^	0.71 ± 0.13 ^b^	0.58 ± 0.16 ^b^	1188.54 ± 98.46 ^b^	0.29 ± 0.06 ^a^
MPSFD-315	444.51 ± 86.60 ^c^	−0.48 ± 0.13 ^b^	0.53 ± 0.22 ^a^	0.39 ± 0.15 ^a^	789.33 ± 65.15 ^a^	0.27 ± 0.08 ^b^
FD	300.07 ± 56.83 ^a^	−0.46 ± 0.10 ^b^	0.80 ± 0.26 ^c^	0.66 ± 0.21 ^c^	1275.65 ± 81.65 ^c^	0.27 ± 0.09 ^b^

Values with different letters in the same column are significantly different (*p* < 0.05).

**Table 5 foods-12-03437-t005:** Effect of freeze-drying (FD) and different infrared-assisted freeze-drying (IRFD) trials on texture and color change in banana slices.

Drying Methods	Crispness (N/mm)	Hardness (N)	∆E
Raw banana	0.1 ^d^	0.6 ^c^	-
FD	1.6 ^c^	7.9 ^b^	4.6 ^a^
IRFD-2.7 kW/m^2^ at 20% WR	1.8 ^bc^	8.8 ^b^	3.3 ^b^
IRFD-2.7 kW/m^2^ at 20% WR to 4.0 kW/m^2^ at 0 °C	1.9 ^b^	8.7 ^b^	2.5 ^b^
Continuous IRFD-2.7 kW/m^2^	2.8 ^a^	13.8 ^a^	2.4 ^b^

^a–d^ Means (±standard deviation) with a different letter in the same column are significantly different at *p* < 0.05.

**Table 6 foods-12-03437-t006:** Sensory evaluation of samples dried using freeze-drying (FD) and different infrared and microwave-vacuum-assisted freeze-drying (IRFD) trials.

Drying Methods	Appearance	Texture	Mouthfeel	Total Score
FD	2.36 ± 0.21 ^b^	2.43 ± 0.09 ^a^	3.43 ± 0.15 ^a^	8.22 ± 0.29 ^b^
IRFD	2.59 ± 0.18 ^a^	2.52 ± 0.21 ^a^	3.51 ± 0.19 ^a^	8.62 ± 0.26 ^a^
IRFD3h-MVD	1.69 ± 0.17 ^c^	1.51 ± 0.16 ^d^	2.34 ± 0.22 ^d^	5.54 ± 0.27 ^e^
IRFD4h-MVD	2.20 ± 0.18 ^b^	1.91 ± 0.20 ^c^	3.02 ± 0.19 ^c^	7.13 ± 0.21 ^d^
IRFD5h-MVD	2.31 ± 0.13 ^b^	2.11 ± 0.14 ^b^	3.20 ± 0.17 ^b^	7.62 ± 0.18 ^c^

Different letters in the same column indicate significant differences (*p* < 0.05).

**Table 7 foods-12-03437-t007:** Effect of different freeze-drying techniques on some chemical and sensory parameters of button mushrooms [45].

Chemical and Sensory Parameters	FD	FD+MVD	FD+VD	FD+AD
Proteins (mg/g d.b.)	73.92 ± 3.91 ^a^	70.69 ± 7.43 ^ab^	63.46 ± 10.07 ^ab^	52.50 ± 2.60 ^b^
Vitamin C (mg/g d.b.)	2.81 ± 0.11 ^a^	1.63 ± 0.30 ^c^	2.40 ± 0.16 ^ab^	2.05 ± 0.06 b^c^
Reducing sugar (mg/g d.b.)	26.32 ± 0.22 ^a^	23.18 ± 0.10 ^b^	23.78 ± 1.50 ^b^	20.62 ± 0.65 ^c^
Total sugar (mg/g d.b.)	48.07 ± 1.68 ^a^	40.76 ± 3.73 ^ab^	36.32 ± 2.06 ^b^	34.35 ± 4.22 ^b^
Hardness (g)	177.58 ± 44.20 ^a^	211.37 ± 50.23 ^ab^	204.46 ± 37.62 ^ab^	230.43 ± 46.32 ^b^
Fracturability (mm)	16.55 ± 0.42 ^a^	17.53 ± 0.96 ^b^	18.22 ± 0.91 ^bc^	18.60 ± 0.42 ^c^

Values with different lowercase letters are significantly different (*p* < 0.05).

**Table 9 foods-12-03437-t009:** Comparison between the operations and capital costs for a laboratory and an industrial freeze-dryer [63].

Freeze-Dryer	Operational Cost, EUR/Cycle	Capital Cost, EUR/Cycle	Total Cost, EUR/Cycle	Total Cost, EUR/Dose
Laboratory	3.28	29.76	33.04	0.041
Industrial	107.27	595.24	702.51	0.007

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
