# Peer review of "Recent Developments in the Hybridization of the Freeze-Drying Technique in Food Dehydration: A Review on Chemical and Sensory Qualities"

_foods, 2023, doi:10.3390/foods12183437_

Round 1
Reviewer 1 Report
Overall, writing need to be revised and digest the information. Language should be more sharp as a quality manuscript.
is this a hybridization of freeze drying or hybridization of drying technique?
There are many errors in the texts including unexpected bold (line 32,221, line 33, p14,), font ( line 32), line numbering, picture resolution are very poor.
Line 39: How freeze dry retain bioactive molecules and how DNA belong to bioactive molecules: which compound is antibiotic in plant? what is sensitive material in plant list down please?
line 48: is protein more degradative material? what is the degradation temperature of protein? what does microbial action mean?
is freeze only cover fruits, vegetables and herbs?
line 393: words are not unified. there are many more line where words are not unified.
line 419: if single freeze dry increase the bioactive content from 3 to 5 times compared to FD EFD, so why and how this technique can be align with the industrial process.
line 421: 'Hence, the FD-EPD method is a potent drying technique that could be used to turn pumpkin slices into snack foods like chips while maintaining their functional and textural properties with less energy'? how this statement can be made? regarding line 419.
line 467: Applications of hybrid freeze-drying techniques in food dehydration: why only IRFD and MFD are being discussed in this section. all related technique should be discussed under this heading.
In conclusion: how this statement: Freeze-drying is a well-accepted drying technology is being made since this technology is tested in lab scale only?
Extensive editing of English language required
Reviewer 2 Report
In this review, the authors discuss the Recent Developments in the Hybridization of Freeze-Drying Technique in Food Dehydration regarding their Chemical and Sensory Qualities. The topic is interesting and the manuscript could be have potential from a scientific point of view. However, there are some flaws that require the authors’ attention.
1. Introduction: This section presents rather general information regarding the Freeze-drying technique and its uses. I believe a small table presenting all pros and cons of freeze-drying (as well as its uses) would be really helpful for the reader.
2. Also, paragraphs 1 and 2 could be better organized and present first FD uses, then the physicochemical factors, then effect of the technique on foods, etc.
3. Lines 89-102: I believe the authors intended to use this paragraph as a startpoint for presenting the hybrid techniques (later on). A few lines talking about hybrid strategies and their advantages in general should be enough. Lines 96-102 seem to have no use and could be completely omitted. Please edit the whole paragraph accordingly.
4. Hybrid subsections: All subsections (lines 104-466), despite informative, contain parts that are too technical to understand or confusing. After each technique the authors could include a small table (like table 1) presenting the effects of each different freeze-drying technique. Most importantly, I believe a small paragraph referring directly to the effect of each FD technique on Chemical and Sensory Qualities of Foods is important (at the end of each freeze-drying variation subsection). After all, this is the title of the manuscript.
5. Table 2: This table is rather good. I would suggest the authors to emphasize on Chemical and Sensory Qualities of each food (in separate columns for clarity reasons). Another column could also be added presenting this time data about the comparisons of each technique with simple freeze-drying method (if available). Alternatively, this information could be added in-text (lines 468-496).
6. Subsection “Benefits of a hybrid freeze-drying system in food drying” : It would be interesting to see economical data (except for percentages), if available, per ton or piece of product.
7. Conclusions paragraph should be improved. Start with a summary (not too technical), and finalize with an outlook. More precise and comprehensive conclusions can be provided for readers. At its current state this section is rather poor, but the scientific overview given in the manuscript is more complex.
Moderate changes required.
Round 2
Reviewer 1 Report
The authors carefully revised the corrections and improved the quality of the manuscript. I have found a few issues which need to be fixed.
Line 63: Not breaking down the phenolic compounds ' should be breaking down the cell wall to release the phenolic compounds'.
Line 191: should be 'total phenols'
Figures 2 and 3: Please improve the quality of the image. still very poor quality.
Table 6: Please correct the top/second border
Table 8/9: Please unify the top border width
all are OK.
Minor editing of English language required
Author Response
Response to Reviewer 1 Comments
- The authors carefully revised the corrections and improved the quality of the manuscript. I have found a few issues which need to be fixed.
Line 63: Not breaking down the phenolic compounds ' should be breaking down the cell wall to release the phenolic compounds'.
♣ The sentence was corrected as follows: "Further, the freezing process results in the breakdown of cell wall and the release of bound phenolics".
Line 191: should be 'total phenols'
♣ Corrected.
Figures 2 and 3: Please improve the quality of the image. Still, they have very poor quality.
The quality of Figures1, 2, and 3 was improved.
Table 6: Please correct the top/second border.
♣ Corrected.
Table 8/9: Please unify the top border width.
♣ Unified.
- Minor editing of English language required.
♣ The paper was double checked to improve the English language level.

Reviewer 2 Report
Authors have addressed my comments
Minor editing
Author Response
Response to Reviewer 2 Comments
- Minor editing of English language of the paper.
♣ The paper was double checked to improve the English language level.
